# A Rice Hazards Risk Assessment Method for a Rice Processing Chain Based on a Multidimensional Trapezoidal Cloud Model

**DOI:** 10.3390/foods12061203

**Published:** 2023-03-12

**Authors:** Jiabin Yu, Huimin Chen, Xin Zhang, Xiaoyu Cui, Zhiyao Zhao

**Affiliations:** 1School of Artificial Intelligence, Beijing Technology and Business University, Beijing 100048, China; yujiabin@th.btbu.edu.cn (J.Y.); zhangxin@btbu.edu.cn (X.Z.); xiaoyucui@btbu.edu.cn (X.C.); 2Key Laboratory of Industry Industrial Internet and Big Data, China National Light Industry, Beijing Technology and Business University, Beijing 100048, China

**Keywords:** rice processing chain, risk assessment, multidimensional trapezoidal cloud model, dynamic weight

## Abstract

Rice is common in the human diet, making rice safety issues important. Moreover, rice processing safety is key for rice security, so rice processing chain risk assessment is critical. However, methods proposed to assess the rice processing chain risk have issues, such as the use of unreasonable thresholds for the rice processing chain and fixed weight. To solve these problems, we propose a risk assessment method for the rice processing chain based on a multidimensional trapezoidal cloud model. First, an evaluation model based on a multidimensional trapezoidal cloud model was established. Based on the historical evaluation results, Atanassov’s interval-valued intuition language numbers (AIVILNs) were introduced to determine the cloud model’s parameters. Second, the concept of dynamic weight was introduced to integrate the static and dynamic weights. An exponential function was used to construct dynamic weighting mechanisms, and the analytic hierarchy stage (AHP) was used to construct a static weight. The proposed method was validated by 104 sets of rice processing chain data, and the results show that the method could evaluate the risk level of the rice processing chain more accurately and reasonably than other methods, indicating that it can provide a sound decision-making basis for food safety supervision authorities.

## 1. Introduction

In China, a major rice producer, more than 60% of the population consumes rice as a staple food. In recent years, however, food safety incidents have occurred frequently, such as cadmium and mildew contaminations in rice. These incidents have raised concerns about rice safety and demonstrated the need for stricter requirements for rice safety supervision. Rice processing safety is an important basis for rice security, so it critical for evaluating the security risk of the rice processing chain.

Currently, food risk assessment methods include qualitative assessment methods, quantitative assessment methods, and qualitative–quantitative comprehensive risk assessment methods [1]. Qualitative assessment methods are mainly based on the knowledge and experience of the expert to analyze risk indexes [2,3,4], including the index scoring method [5], Delphi method [6], and hazard analysis and critical point control (HACCP) method [7]. For example, Du et al. [8] investigated a weighting method based on an expert knowledge structure analysis, which effectively considered the opinions of various experts and improved the consensus of the cluster. However, the qualitative evaluation method relies too much on subjective factors and cannot be used to accurately construct an early warning model.

Quantitative assessment methods are data-driven models [9]. They are based on data to establish a mathematical model and use the mathematical model to calculate the risk value of the index [10]; these models include the random forest algorithm [11], support vector machine (SVM) [12], and back-propagation (BP) network [13]. For example, Fang et al. [14] combined the red cabbage anthocyanin label with a BP neural network to propose a smart phone application, forming a simple system that could quickly scan the label and identify the freshness of fish in real time. However, quantitative evaluation methods have problems, such as an overdependence on data.

The qualitative and quantitative comprehensive analysis method is the combination of qualitative and quantitative evaluation methods [15], and it is the focus of current research. These methods include the analytic hierarchy stage (AHP) [16], fuzzy comprehensive evaluation [17], and cloud model [18]. AHP is a method that builds a risk judgment matrix and combines it with the hierarchical structure model through the establishment of expert opinions. It essentially quantifies the empirical judgments of decision makers but lacks responsiveness to actual data. Fuzzy comprehensive evaluation can be applied to the analysis and evaluation of both subjective and objective indexes, and it can describe the fuzzy features in the evaluation problem well [19]. The cloud model can describe the randomness, fuzziness, and correlation of concepts in natural language and then transform the quantitative risk value and interval with the qualitative language set according to the membership degree theory. It is widely used in risk assessment. At present, the normal cloud model is the most commonly used method. Xu et al. [20] used the normal cloud model to evaluate the quality of red wine and realized the conversion between quantitative data and the qualitative evaluation level. However, there are still some limitations in the practical application. First, the normal cloud model uses a numerical point to represent an evaluation level. In terms of actual production risk assessment, the risk level is a numerical interval, which can have a certain impact on the evaluation results. Second, weighting methods are usually used to obtain a weight for a single dimension, which leads to inaccurate weight acquisition. Finally, the classification of risk levels in the current research is mostly based on national standards, which are put forward for general situations. However, the conditions of different processing chains vary, and the threshold interval of the national standard may introduce errors into the values of model parameters, resulting in inaccurate evaluation results. Table 1 shows a comparison of the advantages and disadvantages of various risk evaluation methods.

Some scholars have conducted research on the above problems. In terms of actual production risk assessment, the evaluation level of the trapezoidal cloud model is a numerical interval, so a model is used for risk assessment. Zhang et al. [21] applied the trapezoidal cloud model to the risk assessment of subway fire accidents, revealing the superiority of the trapezoidal cloud model. Wang et al. [22] used asymmetric trapezoidal cloud (ATC) models to analyze an illustrative case study involving the evaluation of industrial sewage discharge. The current weighting methods include AHP, grey correlation analysis, and the coefficient of variation method. To establish a successful energy cooperation, Papapostolou et al. [23] combined AHP with the fuzzy technique for order of preference by similarity to ideal solution (Fuzzy TOPSIS) methods for adopting the most appropriate strategic plan. To combine subjective and objective weights, Niu et al. [24] proposed a method combining AHP and information entropy weights to evaluate and predict the health status of production lines. However, the above weighting methods are static, and the results obtained after weight acquisition are fixed values, which cannot be changed according to changes in production conditions. Xin et al. [25] proposed a dynamic weight mechanism and introduced a variable weight coefficient to replace the constant weight with a variable weight. Zhao et al. [26] used an exponential function to model the dynamic weight mechanism. However, the above variable weight coefficient and the base number of exponential functions are obtained from expert experience, which is too subjective. Finally, Qian et al. [27] introduced Atanassov’s interval-valued intuitionistic fuzzy sets (AIVIFS) to improve the ladder cloud model, and they gave an example for the grade evaluation of COVID-19. However, AIVIFS are limited to describing quantitative information, and in the case of quantitative expression, their description may be vague, which is inconvenient for applications. Therefore, it is more appropriate to provide estimates by a verbal value than a numerical value. Yu et al. [28] applied AIVILNs to improve the trapezoidal cloud model to evaluate the eutrophication of a specific water environment.

Based on the above literature, two problems have yet to be solved. First, most of the current weighting methods are static, and the results obtained after weight acquisition are fixed values, which cannot be changed according to changes in production conditions. Moreover, the variable weight coefficient and dynamic weight parameters of the dynamic weight methods are obtained by experts’ experience, which is too subjective. Second, existing research is restricted to describing quantitative information, and in the case of referring to quantitative expression, the descriptions made by them can be ill-defined, creating inconvenience in application. Therefore, it is more suitable to provide assessments by means of linguistic values rather than numerical values.

This paper presents a risk assessment of rice processing chain hazards based on a multidimensional trapezoidal cloud model. The proposed method was validated by processing chain data.

The main work of this paper is as follows:

(1) The multidimensional trapezoidal cloud model is used for the risk assessment of a rice processing chain to realize the transformation between quantitative hazard data and the qualitative evaluation level;

(2) AIVILNs are introduced to determine the model parameters, and the digital feature determination method of the trapezoid cloud model is improved to overcome the unreliable evaluation results and low identification caused by the inherent national standard classification threshold in the trapezoid cloud model;

(3) The concept of dynamic weight is introduced, and the dynamic weight mechanism is modeled by using an exponential function model. The constant weight is replaced with variable weights, making the weight determination more reliable.

## 2. Materials and Methods

### 2.1. Data Characteristics and Processing

In this paper, the rice hazards data in five rice producing areas in China in 2022 were used as an example for analysis. This paper uses the rice hazards data in the processing chain at four stages, namely, paddy rice, husking, polishing, and polished rice for the example analysis. The data include the rice origin, detection time, and detection elements and results. Initially, the data are multidimensional, and many attributes, such as rice varieties, are redundant in determining the degree of risk, so it is necessary to screen the data before risk assessment. The data of the rice hazards are not equivalent in regard to the order of magnitude, which is not conducive to the construction of subsequent risk assessment models, so the indexes must be standardized. The rice hazards all belong to the negative effect class, which means that the index detection value must be lower than the limit value. Therefore, reverse index standardization of the data is required.

First, the metrics are screened. The types of rice hazards include heavy metals and mycotoxins. It has been shown that heavy metals easily accumulate in rice grains, leading to excessive heavy metal content in rice [29], which then enters the human body through the diet, leading to a variety of diseases and threatening human health. Mercury, lead, and chromium, all common heavy metal pollutants in rice, were selected as heavy metal rice hazards. Mycotoxins are secondary metabolites produced by fungi, among which aflatoxin B_1_ (AFB_1_) and zealenone (ZEA) are common mycotoxins in rice and are highly toxic to both humans and animals [30]. These five indexes widely exist in real production, do strong harm, and are easier to detect than other factors. Therefore, these five indexes are selected as risk indexes to judge the safety degree of the rice processing chain.

Second, the purpose of data processing is to eliminate the difference in dimensions and magnitudes of the indexes so that the indexes are comparable. We adopt reverse index standardization, given by
(1)$$x_{ij}=\frac{\max_{i}X_{ij}-X_{ij}}{\max_{i}X_{ij}-\min_{i}X_{ij}}$$
where *X_ij_* is the original value of risk index *I* in processing chain *j*. A sample of rice hazard testing data is shown in Table 2.

### 2.2. Risk Assessment Method of a Rice Processing Chain Based on a Multidimensional Trapezoidal Cloud Model

According to the index screening, five factors, that is, ZEA, AFB_1_, mercury, lead, and chromium, were adopted as the main indexes of the risk assessment. Thus, a five-dimensional trapezoidal cloud model was constructed. First, the AIVILNs are used to determine the parameters of cloud model. According to the rice hazard data, the expectation, upper and lower limits of the expectation, entropy, and excess entropy of the multidimensional trapezoidal cloud model are determined by the AIVILNs. Second, the comprehensive weights of the risk indexes are obtained through the method of combining static and dynamic weights. Finally, the above model parameters, risk index weights, and rice hazard data are all input into the multidimensional trapezoidal cloud model to obtain the corresponding level of the membership degree. The final evaluation result is the level with the maximum membership degree. The specific stage is shown in Figure 1.

#### 2.2.1. Construction of Risk Assessment Index System for Rice Hazards

Based on the China national standard [31], we construct a rice safety risk assessment standard table. Rice risk is divided into six levels. The quantization value corresponding to each level is ≤0.2*a*, ≤0.4*a*, ≤0.6*a*, ≤0.8*a*, and >*a*, where *a* represents the national standard. The standards are shown in Table 3.

#### 2.2.2. Weighting Method

In the risk assessment of a rice processing chain, the weight of the risk indexes should be constantly changed with different stages, but the traditional weighting method cannot dynamically determine the weight of the risk indexes. To describe the weights of different risk indexes in different segments, the weights are divided into static and dynamic weights. Let the static weight be *w*_0_, which represents the fixed weight of each risk index. It represents the subjective weight [32] and is obtained through AHP [33] based on expert opinion. Let the dynamic weight be *w_d_*, which changes constantly according to the different rice hazard data. In this paper, the exponential function is adopted for modeling, and the coefficient of variation method is combined for calculation. Let the combined weight obtained from the combination of the two be *W*, which makes the weight more accurate.

According to the experience gained in actual production, we know that the rice hazard detection data in each stage differ. In other words, the impact on security is also different. The greater the value of rice hazards data, the greater its impact on security, reflected in the corresponding dynamic weight index increase. When the detection value of a certain risk index in a certain stage increases, the dynamic weight index of the risk index in the stage increases, and its increasing trend shows a certain degree of exponential growth. This is shown in Figure 2.

As shown in Figure 2, the horizontal axis represents the risk index detection data, and the vertical axis represents the weighting index. When the risk index detection data value of a certain stage of the rice processing chain is 0, the weight is the static weight *w_0_*. The dynamic weight *w_d_* is modeled by an exponential function, which can be expressed as follows:(2)$${\left(w_{d}\right)}_{ij}=a_{ij}^{x_{ij}^{\prime }}$$
where *x’_ij_* is the normalized result of the impact factor, *a_ij_* is the bottom value of the exponential function of the dynamic weight of risk index *i* in stage *j*, and *a_ij_* > 1. Different hazards have different values, which are determined by the impact of hazard *i* on the security of this stage. The greater the impact, the greater the value. The bottom value of the dynamic weight exponential function represents a kind of deviation, which is the degree of deviation of the detection value from 0. The greater the difference between the detection values of risk index *i*, the greater the value of *a_ij_* should be. Therefore, the coefficient of the variation method is adopted to obtain the value. The coefficient of variation method is an objective weighting method. The index with a greater difference in the evaluation index system can better reflect the gap of the risk index, and the weight of such an index is larger in the index system. The dynamic weight is calculated as follows:(3)$${\left(A_{j}\right)}_{i}=\frac{1}{n}{\sum_{m=1}^{n}\left[{\left(x_{j}\right)}_{i}\right]}_{m}$$
(4)$${\left(S_{j}\right)}_{i}=\sqrt{\frac{1}{n}\sum_{m=1}^{n}{\left\{{\left[{\left(x_{j}\right)}_{i}\right]}_{m}-{\left[{\left(A_{j}\right)}_{i}\right]}_{m}\right\}}^{2}}$$
(5)$${\left(a_{j}\right)}_{i}=\frac{{\left(S_{j}\right)}_{i}}{{\left(A_{j}\right)}_{i}}$$
where *m* is the amount of data. We have *i* = 1, 2, …, 5 in Equations (3)–(5), representing the five risk indexes. The values of *j* in Equations (3)–(5) range from 1 to 4, representing the four stages.

The dynamic weight vector of the indexes is calculated according to Equation (2). Taking paddy rice as an example, we set the dynamic weight of paddy rice as *w_d_*_1_. The weight vector of paddy rice is as follows: *w_d_*_1_ = (0.1984, 0.2007, 0.1938, 0.1953, 0.2117).

To verify the effectiveness of dynamic weight, we display several sets of data, and the dynamic weights of these data are calculated. It is shown that when the data change, the dynamic weight vector also changes, as shown in Table 4.

In Table 4, the sampling data are the detection data of the risk index, and the meanings of each data set are in the same order as the list of parameter names in Table 3. For example, the first data set is ZEA sampling data, and the second data set is AFB_1_ sampling data. The weight *W* is the comprehensive weight obtained by combining static and dynamic weights. Through the comprehensive weight, we can intuitively see the relative proportion of current risk indexes. It can be seen from the above table that when the rice hazard data change, the corresponding weight also changes dynamically. The parameter ZEA in data sets 1 and 2 changes significantly. When the data of risk index ZEA decreases, the weight of ZEA also decreases. The parameters of AFB_1_ and mercury in data sets 3 and 4 vary greatly. As their data increase, their weights also increase. In data sets 5, 6, 7, and 8, three parameters show obvious changes. It can be seen that when the parameter data sets are larger, the weight coefficient increases, and when the parameter data sets are smaller, the weight coefficient decreases.

The comprehensive weight *W* is obtained according to static weight and dynamic weight. The equation for the comprehensive weight is as follows:(6)$$W=\alpha w_{0}+\beta w_{d}$$

It must satisfy the constraint conditions of unification: (7)$$\alpha^{2}+\beta^{2}=1,\alpha \geq 0,\beta \geq 0$$

The values of *α* and *β* can be determined subjectively by the decision maker or obtained by consulting experts. According to [34], we set *α* = 0.5 and *β* = 0.5.

In this paper, the static weight is obtained by the AHP. According to the hierarchical structure of the index system, the index weight is calculated by using the 1–9 scale method scored by experts [35]. Taking paddy rice as an example, we determine that the static weight of paddy rice is *w_0_* = (0.4757, 0.2856, 0.1377, 0.0314, 0.0695).

According to Equation (6), the comprehensive weight of paddy rice can be calculated as *W* = (0.3635, 0.1769, 0.0988, 0.0465, 0.3144).

#### 2.2.3. Construction of the Evaluation Model of the Processing Chain

The cloud model is a model based on stochastic mathematics and fuzzy mathematics theory, which is used to deal with the conversion of quantitative data and qualitative concepts [36]. In this paper, a multidimensional trapezoidal cloud model is used to assess the security risk of a rice processing chain.

The trapezoidal cloud model can be expressed as $\underline{E_{x}},\bar{E_{x}},E_{n},H_{e}$. $\underline{E_{x}}$ is the lower limit of the expectation, $\bar{E_{x}}$ is the upper limit of the expectation, *E_n_* is the entropy, and *H_e_* is the excess entropy.

According to Table 3, rice risk levels are divided into six levels. In a certain stage, the membership degree of each level corresponding to the risk index is calculated according to the stage.

According to the literature [37], the membership degree *μ* is as follows:(8)$$\mu =\exp\left[-f\frac{{\left(x-E_{x}\right)}^{2}}{2{\left(E_{nn}\right)}^{2}}\right]$$
where *E_nn_* is a normal random number generated by *E_n_* and *H_e_*, and *f* is the comprehensive weight of each hazard index. We have *i* = 1, 2, …, 5, which represents the five risk indexes.
**Method flow**Step 1**if** $\underline{E_{x_{1}}}\leq x_{1}\leq \bar{E_{x_{1}}}\&\&\underline{E_{x_{2}}}\leq x_{2}\leq \bar{E_{x_{2}}}\&\&\ldots \&\&\underline{E_{x_{5}}}\leq x_{5}\leq \bar{E_{x_{5}}}$Step 2*μ* = 1Step 3**else**Step 4**  for** *i* = 1:5Step 5    **if** $\underline{E_{x_{i}}}\leq x_{i}\leq \bar{E_{x_{i}}}$Step 6     $E_{x_{i}}=E_{x_{i}}$
Step 7    **else if** $x_{i}>\bar{E_{x_{i}}}$
Step 8     $E_{x_{i}}=\bar{E_{x_{i}}}$
Step 9    **Else**Step 10     $E_{x_{i}}=\underline{E_{x_{i}}}$
Step 11    **End**Step 12  **End**Step 13$\mu =\exp\left[-\sum_{i=1}^{5}\left(f_{i}\frac{{\left(x-E_{x_{i}}\right)}^{2}}{2{\left(E_{nn_{i}}\right)}^{2}}\right)\right]$Step 14**End**


Through the above process, a multidimensional trapezoidal cloud model is constructed to obtain the membership degree of influence factors in different stages at different levels. The evaluation result is the level with the maximum membership degree.

#### 2.2.4. Construction of AIVILNs and the Evaluation Model Parameter Calculation

In traditional trapezoidal cloud model parameters, the upper limit $\bar{E_{x}}$ and lower limit of expectation $\underline{E_{x}}$ are determined based on the limits of evaluation standards. The threshold interval is based on the national standard, as shown in Table 3. Typically, the national standard is wider than the data and is more prone to errors.

AIVILNs express the membership degree and non-membership degree in the form of an interval value, which can describe the uncertainty of qualitative information and improve the above problems. Therefore, this paper adopts AIVILNs to determine the parameters of the multidimensional trapezoidal cloud.

The AIVILNs are defined as follows: $A=\{<X,h_{j},[a,b],[c,d]>\}$, $h_{j}\in H$, [a,b]⊆[0,1], [c,d]⊆[0,1], and b+d≤1, where *a*, *b*, *c*, *d* can be acquired from the membership degrees of the historical evaluation results. [a,b] indicates the membership degree belonging to a level, and [c,d] represents the membership degree of the same index that does not belong to a certain level. After index screening, the five evaluation indexes of ZEA, AFB1, mercury, lead, and chromium are selected as the basis of this evaluation. An influencing factor set $X=\left\{X_{1},X_{2},\ldots ,X_{5}\right\}$ and a language item set $H=\left\{h_{j}|j=1,\cdots ,2n,n\in N^{*}\right\}$ are constructed where *h_j_* represents the rice safety level. Because there are six levels in Table 3, the value of *j* ranges from 1 to 6. The language item set of the rice safety level in this paper is constructed as $H=\{h_{1},h_{2},h_{3},h_{4},h_{5},h_{6}\}$. The AIVILNs are implemented as follows.

The numerical parameter *θ_j_* is mapped from the language item set of the rice safety level. The formula is as follows:(9)$$\theta_{j}=\{\begin{array}{cc} \left(a^{t}-a^{t-j})/(2a^{t}-2\right) & \left(0<j\leq t\right)\\ \left(a^{t}+a^{j-t}-2)/(2a^{t}-2\right) & \left(t<j\leq 2t\right) \end{array}$$
where *t* = 3. For the same processing chain, the parameter a in this equation is fixed. According to the literature [38], the general range of a is [1.36, 1.4].

The formulas for calculating parameters of the multidimensional trapezoidal cloud model are as follows [39]:

The parameters $E_{x_{j}}$, $\bar{E_{x_{j}}}$, and $\underline{E_{x_{j}}}$ of the trapezoidal cloud model are as follows:(10)$$E_{x}{}_{{}_{j}}=X_{\min}+\theta_{j}(X_{\max}-X_{\min})$$
(11)$$\underline{E_{x_{j}}}=[\frac{1}{2}+\frac{1}{4}(a+b+c+d)]E_{x_{j}}$$
(12)$$\bar{E_{x_{j}}}=[\frac{3}{2}-\frac{1}{4}(a+b+c+d)]E_{x_{j}}$$
where *X_max_* and *X_min_* are the maximum and minimum values of the measured indexes, respectively.

The parameters *E_nj_* and *H_ej_* of the trapezoidal cloud model are as follows:(13)$$\max\{E_{n}{}_{{}_{j}}\}=\frac{\max\{X_{\max}-E_{x}{}_{{}_{j}},E_{x}{}_{{}_{j}}-X_{\min}\}}{3}$$
(14)$$\min\{E_{n}{}_{{}_{j}}\}=\frac{\max\{X_{\max}-\frac{3}{2}E_{x}{}_{{}_{j}},\frac{1}{2}E_{x}{}_{{}_{j}}-X_{\min}\}}{3}$$
(15)$$E_{n}{}_{{}_{j}}=\frac{\max\{E_{n}{}_{{}_{j}}\}+\min\{E_{n}{}_{{}_{j}}\}}{3}$$
(16)$$H_{e}{}_{{}_{j}}=\frac{\max\{E_{n}{}_{{}_{j}}\}-E_{n}{}_{{}_{j}}}{3}$$
where *E_nj_* and *H_ej_* are the entropy and excess entropy of each level, respectively.

## 3. Experiments and Results

The experimental environment was a Windows 11, 64-bit operating system, the processor used was an AMD R7-5800H CPU @ 3.2 GHz, the running memory was 32 GB, the graphics card was RTX 3050Ti, and the experiment was based on MATLAB R2021b. The proposed method was validated by 104 sets of rice processing chain data. The experiment was conducted based on this environment configuration.

### 3.1. Risk Safety Evaluation in the Rice Processing Chain

First, according to the comprehensive weight calculation method, the comprehensive weights of the risk indexes in the four stages of the rice processing chain were calculated by applying Equations (2)–(6). The combined weight calculation results are shown in Table 5.

Based on Equations (9)–(16) in Section 2.2.4, the AIVILNs of each level are shown in Table 6.

Multidimensional trapezoidal cloud model parameters were calculated according to AIVILNs and are shown in Table 7.

The rice hazard data in the rice processing chain and the parameters of the multidimensional trapezoidal cloud calculated by the AIVILNs were all input into the trapezoidal cloud model. Combining the weight of each index in Table 5, we calculated the membership degrees for each level according to the method flow. The values are shown in Table 8.

In Table 8, it can be seen that paddy rice had the maximum membership degree in level III, so the level of paddy rice is defined as level III. Similarly, it can be seen that the stages of husking, polishing, and polished rice were all defined as level I.

### 3.2. Comparison Experiments

To verify the effectiveness of the proposed method, we next compared three algorithms. Algorithm 1 uses a traditional trapezoidal cloud and static weight (without AIVILNs and dynamic weight). Algorithm 2 uses a traditional trapezoidal cloud and dynamic weight (without the AIVILNs). The algorithm of the proposed method uses AIVILNs and dynamic weight. The membership degree of each level and the evaluation results of different algorithms are shown in Table 9, Table 10, Table 11 and Table 12.

First, by comparing the levels and evaluation results of Algorithm 1 with the algorithm of the proposed method, we can see from Table 9 that the risk level obtained by Algorithm 1 in the stage of paddy rice was level IV, while that obtained by the proposed method’s algorithm was level III. The risk level obtained by Algorithm 1 in the stages of husking and polishing was level II. It can be seen from Table 10 that the risk assessment level obtained by the proposed method’s algorithm was level I. It can be seen from Table 9, Table 10, Table 11 and Table 12 that the weight defect method combined with the dynamic weight adopted in this paper improved the identification degree of the evaluation algorithm and made the weight distribution more reliable.

To further verify the effectiveness of the proposed method’s algorithm, we compared the bar charts and analyzed each level of membership degrees. Figure 3, Figure 4, Figure 5 and Figure 6 show the membership of each level obtained using the three algorithms. It can be seen more intuitively from Figure 3, Figure 4, Figure 5 and Figure 6 that the numerical difference in the membership degree between adjacent evaluation levels obtained by the proposed method’s algorithm was basically more than 0.03, which was greater than that of the two other algorithms. The adoption of the proposed method‘s algorithm improved the identification of the evaluation algorithm, while the identification of the evaluation results obtained by the two other algorithms was low. The use of AIVILNs to optimize the parameters of the evaluation model solved the defect of the traditional trapezoidal cloud model that evaluates based on the inherent classification threshold. Thus, it can make the evaluation results more reliable and accurate.

## 4. Conclusions

In this paper, a rice processing chain evaluation method based on a multidimensional trapezoidal cloud model was proposed, and the rice hazards data from five rice producing areas in China in 2022 were used as an example for analysis. First, by comparing the membership degrees for each level and simulation results of the methods, we found that Algorithm 2 and the proposed method’s algorithm performed extremely similarly. The weight calculation of the method in this paper was more reliable, so it could provide more accurate risk assessment results. Second, by comparing our method’s algorithm with Algorithm 2, we found that the membership degree between the adjacent evaluation layers obtained by our algorithm was significantly different and it improved the recognition degree of the evaluation model. The analysis of the tables and histograms showed that the proposed method’s algorithm improved the recognition of evaluation results. The results indicate that the proposed method can provide an accurate and efficient decision-making basis for food safety supervision authorities.

In future research, the proposed method can be improved in two aspects. First, geographical and temporal factors will be added into the method to extract the relevant risk rules from the data, which will make the evaluation results more reliable and accurate. Second, in terms of the application of the method, real-time monitoring of risk value changes in all of the stages of the processing chain will help reduce the risk threat at the food source.

## Figures and Tables

**Figure 1 foods-12-01203-f001:**
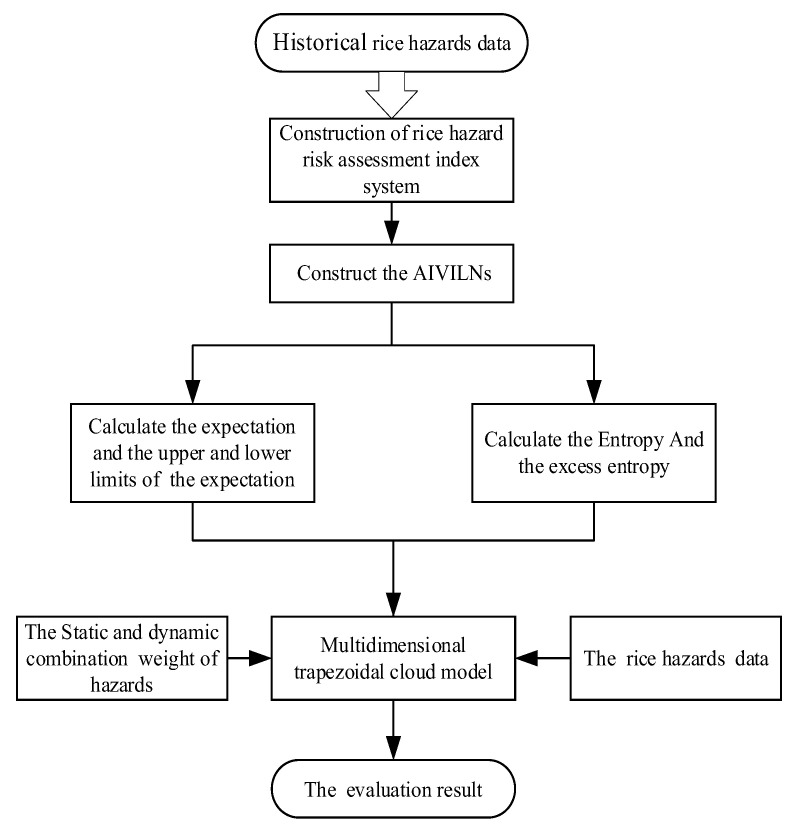
Proposed method flowchart.

**Figure 2 foods-12-01203-f002:**
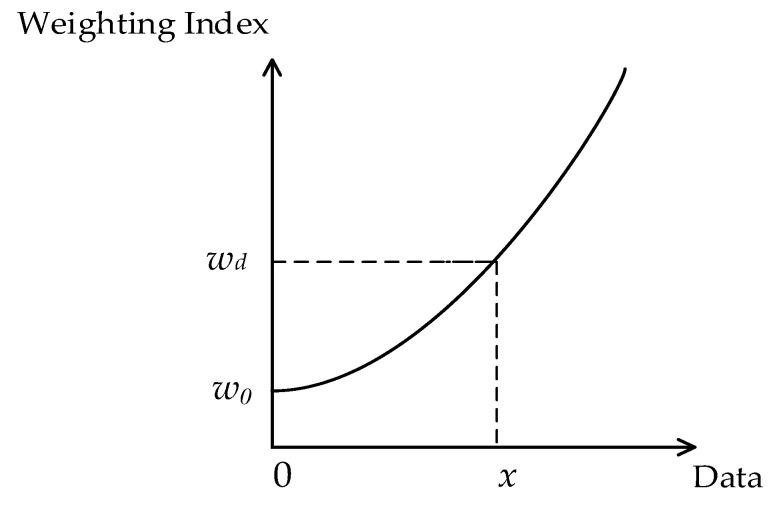
Dynamic weight change pattern.

**Figure 3 foods-12-01203-f003:**
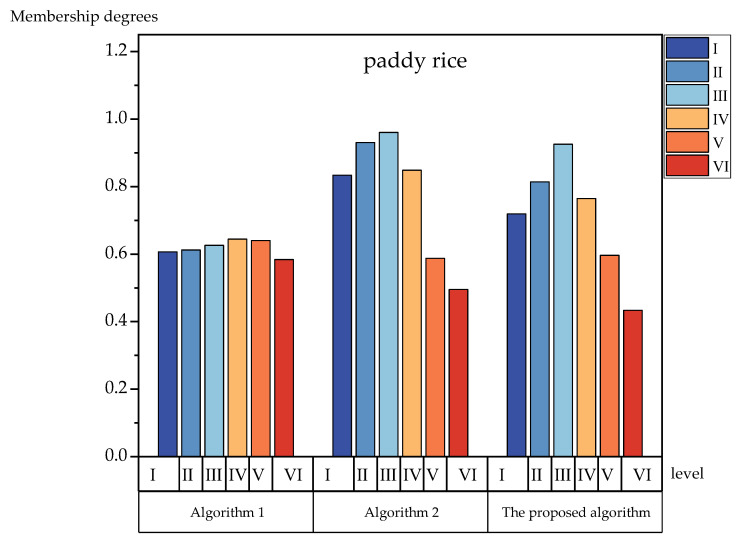
Histograms of the membership degrees of polished rice.

**Figure 4 foods-12-01203-f004:**
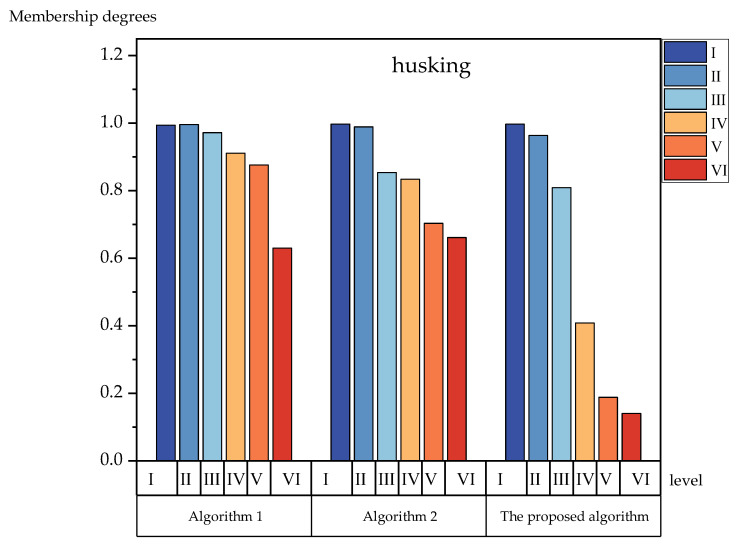
Histograms of the membership degrees of husking.

**Figure 5 foods-12-01203-f005:**
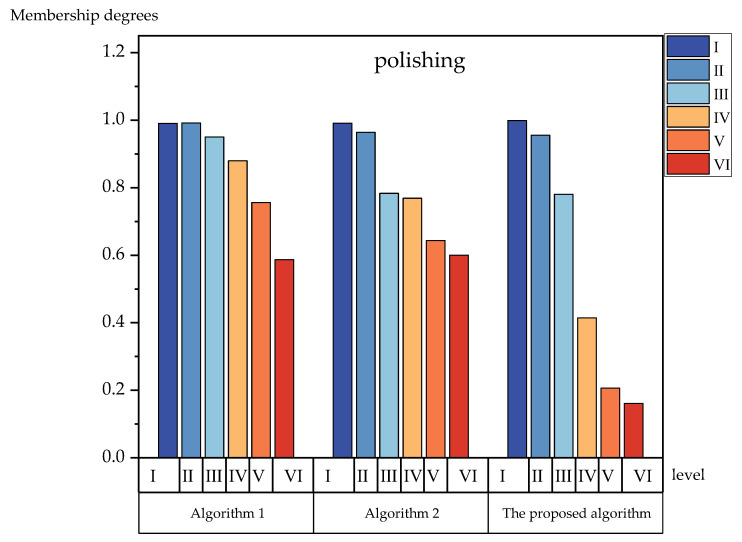
Histograms of the membership degrees of polishing.

**Figure 6 foods-12-01203-f006:**
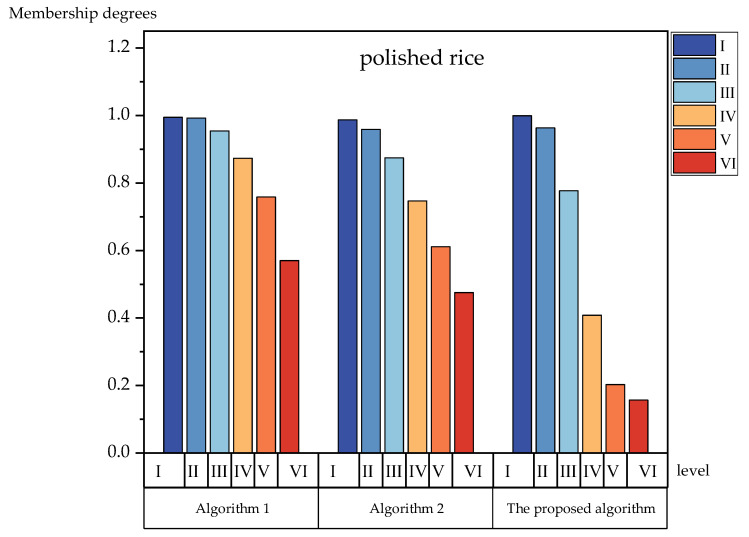
Histograms of the membership degrees of polished rice.

**Table 1 foods-12-01203-t001:** Comparison of advantages and disadvantages of various risk evaluation methods.

Types of Evaluation Methods	Risk Assessment Methodology	Advantages	Disadvantages
Qualitative assessment methods	Index scoring method [5]	Clear quantitative metrics	Difficult to define indicator weights
	Delphi [6]	Relatively simplified relationships between system elements	Complex and time-consuming for collecting expert opinions
	HACCP [6,7]	Multilevel and multi-indicator evaluation	Complex implementation
Quantitative assessment methods	Random forest algorithm [11]	Simple calculation	Prone to overfitting
	SVM [12]	High generalization ability	Unsuitable for classification of large data sets
	BP [13]	High nonlinear mapping capability	Prone to local miniaturization problems
Qualitative and quantitative comprehensive analysis method	AHP [16]	A clear hierarchy of indicators and a wide range of applications	Reliance on the accuracy of expert assessment results
	Fuzzy integrated evaluation [17]	Excellent evaluation results for fuzzy objects	Complex calculation and subjective determination of weights
	Cloud model [18]	Enables conversion of quantitative risk values to qualitative language sets	Difficulty in determining numerical characteristics

**Table 2 foods-12-01203-t002:** Sample of rice hazards data.

Province	Stage	Sampling Date	Hazards				
			ZEA (μg/kg)	AFB_1_ (μg/kg)	Mercury (mg/kg)	Lead (mg/kg)	Chromium (mg/kg)
Anhui	Paddy rice	20210304	30.756	0.671	0.009	0.137	1.300
Anhui	Husking	20210313	0.857	0.143	0.007	0.057	0.495
Jiangsu	Paddy rice	20210411	32.149	0.678	0.059	0.126	7.860
Jiangsu	Polished rice	20210413	0.517	0.200	0.008	0.051	0.156
Heilongjiang	Polishing	20211006	0.640	0.235	0.005	0.050	0.188
Heilongjiang	Polished rice	20211009	0.361	0.187	0.007	0.065	0.269

**Table 3 foods-12-01203-t003:** Rice risk evaluation standards.

Level	Evaluation Parameters				
	ZEA (μg/kg)	AFB_1_ (μg/kg)	Mercury (mg/kg)	Lead (mg/kg)	Chromium (mg/kg)
I	≤12.0	≤2.0	≤0.004	≤0.04	≤0.2
II	≤24.0	≤4.0	≤0.008	≤0.08	≤0.4
III	≤36.0	≤6.0	≤0.012	≤0.12	≤0.6
IV	≤48.0	≤8.0	≤0.016	≤0.16	≤0.8
V	≤60.0	≤10.0	≤0.02	≤0.2	≤1.0
VI	>64.0	>10.0	>0.02	>0.2	>1.0

**Table 4 foods-12-01203-t004:** Part of the data and comprehensive weight vector.

Number	Data	Comprehensive Weight Vector	Stage
	(ZEA, AFB_1_, Mercury, Lead, Chromium) *	(ZEA, AFB_1_, Mercury, Lead, Chromium) *	
1	(29.581, 0.691, 0.016, 0.159, 5.634)	(0.3635, 0.1796, 0.0988, 0.0437, 0.3414)	paddy rice
2	(15.231, 0.724, 0.009, 0.19, 6.112)	(0.3008, 0.1773, 0.0989, 0.0469, 0.3671)	paddy rice
3	(6.17, 0.577, 0.008, 0.074, 0.571)	(0.4251, 0.2176, 0.1341, 0.0817, 0.1145)	husking
4	(5.878, 3.514, 0.608, 0.048, 0.498)	(0.4176, 0.2761, 0.1286, 0.074, 0.1037)	husking
5	(0.713, 0.212, 0.008, 0.055, 0.513)	(0.3458, 0.2423, 0.1622, 0.1095, 0.1402)	polishing
6	(0.681, 3.179, 0.007, 1.352, 2.473)	(0.3075, 0.3004, 0.1295, 0.0851, 0.1775)	polishing
7	(0.512, 0.19, 0.007, 0.047, 0.226)	(0.3475, 0.2424, 0.1637, 0.1114, 0.1351)	polished rice
8	(0.113, 1.89, 0.007, 0.832, 0.226)	(0.3258, 0.2805, 0.1541, 0.1147, 0.1249)	polished rice

* Data sets are arranged in this order.

**Table 5 foods-12-01203-t005:** Combined weights.

Stage	ZEA (μg/kg)	AFB_1_ (μg/kg)	Mercury (mg/kg)	Lead (mg/kg)	Chromium (mg/kg)
Paddy rice	0.3635	0.1769	0.0988	0.0465	0.3144
Husking	0.4521	0.2176	0.1341	0.0817	0.1145
Polishing	0.3458	0.2423	0.1622	0.1095	0.1402
Polished rice	0.3475	0.2624	0.7771	0.1114	0.1351

**Table 6 foods-12-01203-t006:** AIVILNs of each level.

Level	AIVILNs			
	Paddy Rice	Husking	Polishing	Polished Rice
I	$<h_{1},[0.74,0.75],[0.1,0.24]>$	$<h_{1},[0.97,0.98],[0,0.02]>$	$<h_{1},[0.96,0.98],[0,0.02]>$	$<h_{1},[0.96,0.98],[0,0.02]>$
II	$<h_{2},[0.88,0.89],[0,0.11]>$	$<h_{2},[0.94,0.95],[0,0.05]>$	$<h_{2},[0.93,0.95],[0,0.05]>$	$<h_{2},[0.94,0.95],[0,0.05]>$
III	$<h_{3},[0.96,0.97],[0,0.03]>$	$<h_{3},[0.91,0.92],[0,0.08]>$	$<h_{3},[0.78,0.79],[0,0.11]>$	$<h_{3},[0.91,0.92],[0,0.08]>$
IV	$<h_{4},[0.81,0.82],[0.05,0.18]>$	$<h_{4},[0.86,0.87],[0.07,0.13]>$	$<h_{4},[0.66,0.67],[0.12,0.33]>$	$<h_{4},[0.85,0.87],[0.07,0.13]>$
V	$<h_{5},[0.69,0.70],[0.12,0.30]>$	$<h_{5},[0.75,0.76],[0.1,0.24]>$	$<h_{5},[0.62,0.63],[0.22,0.37]>$	$<h_{5},[0.75,0.77],[0.15,0.23]>$
VI	$<h_{6},[0.55,0.56],[0.3,0.44]>$	$<h_{6},[0.51,0.52],[0.3,0.48]>$	$<h_{6},[0.49,0.5],[0.3,0.5]>$	$<h_{6},[0.51,0.52],[0.3,0.48]>$

**Table 7 foods-12-01203-t007:** Multidimensional trapezoidal cloud model parameters.

Level	ZEA	AFB_1_	Mercury	Lead	Chromium
I	(2.6, 2.9, 210/6, 4/18)	(0.43, 0.45, 158/6, 1/18)	(0.00087, 0.00088, 10/6, 0.01)	(0.0087, 0.0088, 6/6, 0.12)	(0.043, 0.045, 30/6, 0.03)
II	(16.56, 16.8, 178/6, 8/18)	(2.76, 2.8, 153/6, 1/18)	(0.0055, 0.0057, 10/6, 0.01)	(0.055, 0.057, 6/6, 0.03)	(0.276, 0.28, 28/6, 0.01)
III	(29.4, 30.6, 145/6, 15/18)	(4.9, 5.1, 147/6, 3/18)	(0.0098, 0.0102, 10/6, 0.002)	(0.098, 0.102, 6/6, 0.03)	(0.49, 0.51, 28/6, 0.01)
IV	(42.6, 44, 112/6, 12/18)	(7.09, 7.34, 142/6, 4/18)	(0.0142, 0.0147, 10/6, 0.03)	(0.142, 0.147, 6/6, 0.02)	(0.71, 0.73, 27/6, 0.02)
V	(55.8, 59.2, 110/6, 9/18)	(9.3, 9.8, 137/6, 2/18)	(0.0186, 0.0197, 10/6, 0.01)	(0.186, 0.197, 6/6, 0.10)	(0.93, 0.99, 27/6, 0.03)
VI	(6, 6.8, 2110/6, 5/18)	(10, 10.6, 135/6, 2/18)	(0.02, 0.028, 10/6, 0.01)	(0.2, 0.28, 6/6, 0.03)	(1, 1.04, 26/6, 0.01)

**Table 8 foods-12-01203-t008:** Membership degrees for each level of the rice processing chain.

Stage	Membership Degrees					
	I	II	III	IV	V	VI
paddy rice	0.7389	0.8137	0.8456	0.7646	0.5965	0.5336
husking	0.9972	0.9635	0.8091	0.4200	0.1882	0.1405
polishing	0.9991	0.9957	0.7807	0.4143	0.2064	0.1611
polished rice	0.9992	0.9632	0.7771	0.4083	0.2028	0.1570

**Table 9 foods-12-01203-t009:** Evaluation results of different algorithms in the paddy rice stage.

Algorithm	I	II	III	IV	V	VI	Evaluation Results
Algorithm 1	0.6067	0.6124	0.6262	0.6443	0.6401	0.5841	IV
Algorithm 2	0.8337	0.9305	0.9606	0.8483	0.5877	0.4951	III
The algorithm of the proposed method	0.7189	0.8137	0.9256	0.7646	0.5965	0.4336	III

**Table 10 foods-12-01203-t010:** Evaluation results of different algorithms in the husking stage.

Algorithm	I	II	III	IV	V	VI	Evaluation Results
Algorithm 1	0.9941	0.9961	0.9722	0.9109	0.8764	0.6298	II
Algorithm 2	0.9975	0.9891	0.8542	0.8340	0.7036	0.6615	I
The algorithm of the proposed method	0.9972	0.9635	0.8091	0.4083	0.1882	0.1405	I

**Table 11 foods-12-01203-t011:** Evaluation results of different algorithms in the polishing stage.

Algorithm	I	II	III	IV	V	VI	Evaluation Results
Algorithm 1	0.9907	0.9919	0.9502	0.8799	0.7563	0.5871	II
Algorithm 2	0.9912	0.9641	0.7833	0.7690	0.6438	0.6004	I
The algorithm of the proposed method	0.9991	0.9557	0.7807	0.4143	0.2064	0.1611	I

**Table 12 foods-12-01203-t012:** Evaluation results of different algorithms in the polished rice stage.

Algorithm	I	II	III	IV	V	VI	Evaluation Results
Algorithm 1	0.9946	0.9923	0.9541	0.8735	0.7586	0.5703	I
Algorithm 2	0.9871	0.9592	0.8746	0.7468	0.6111	0.4753	I
The algorithm of the proposed method	0.9992	0.9632	0.7771	0.4083	0.2028	0.1570	I

## Data Availability

Data is contained within the article.

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
