# Peer review of "A Rice Hazards Risk Assessment Method for a Rice Processing Chain Based on a Multidimensional Trapezoidal Cloud Model"

_foods, 2023, doi:10.3390/foods12061203_

Round 1
Reviewer 1 Report
Dear Authors,
Thank you for submitting your manuscript to the journal "Foods".
This article is interesting and well-written, but there are some small errors. For example:
- in the first line on page 2, please change ".[9]" to "[9]."
- On page 2, lines 9 and 10, I suggest you to write: "Qualitative and quantitative comprehensive analysis method is the combination of qualitative and quantitative evaluation methods[15], and it is the focus of current research."
Related to formula (8), there exist other probability distributions (Laplace distribution, Kumaraswamy Exponential Model, etc). So, some new references could be added.
Thank you again.
Best wishes,
Referee
Author Response
Thank you for your valuable advice.For your question, my improvements are shown in the attachment.Please see the attachment.

Reviewer 2 Report
It is an excellent research work, which presents relevant results and is within the scope of the journal. But, it needs a little improvement.
you need to present the units for parameters in tables 2 and 5... and others Figures 3, 4, 5, and 6 do not have units or axis names. Correct this, as it is very difficult to interpret them.
In tables 4, 6 and 8 it is difficult to understand what the data set inside each parentheses means. I suggest that you open a line beforehand, in the same tables, and insert the set of unknowns that the following numbers represent. or remake the tables in a more understandable way
Author Response
感谢您的宝贵建议。供您参考,我的改进显示在附件中。请参阅附件。

Reviewer 3 Report
In my opinion, the manuscript entitled: "A rice hazards risk assessment for rice processing chain Based on multidimensional trapezoidal cloud model" is interesting, but there are some mistakes.
The Authors should prepare the manuscript strictly according Foods Journal format. Some tables and figures should be corrected and placed elsewhere. Graphs should have names of axis and units. Symbols - italic! in all text. Results and discussion need more comprehensive explanations.Details of my revision are in the attached file.

Author Response
感谢您的宝贵建议。供您参考,我的改进显示在附件中。请参阅附件。

Round 2
Reviewer 3 Report
I have no further comment.